# Artificial and Natural Sweeteners Biased T1R2/T1R3 Taste Receptors Transactivate Glycosylated Receptors on Cancer Cells to Induce Epithelial–Mesenchymal Transition of Metastatic Phenotype

**DOI:** 10.3390/nu16121840

**Published:** 2024-06-12

**Authors:** Elizabeth Skapinker, Rashelle Aldbai, Emilyn Aucoin, Elizabeth Clarke, Mira Clark, Daniella Ghokasian, Haley Kombargi, Merlin J. Abraham, Yunfan Li, David A. Bunsick, Leili Baghaie, Myron R. Szewczuk

**Affiliations:** 1Faculty of Health Sciences, Queen’s University, Kingston, ON K7L 3N9, Canada; 21ess18@queensu.ca (E.S.); 19ra57@queensu.ca (R.A.); 20enc3@queensu.ca (E.C.); 21dg21@queensu.ca (D.G.); 20hlk@queensu.ca (H.K.); 19mja3@queensu.ca (M.J.A.); 2Department of Biomedical & Molecular Sciences, Queen’s University, Kingston, ON K7L 3N6, Canada; svv1@queensu.ca (D.A.B.); 16lbn1@queensu.ca (L.B.); 3Faculty of Science, Biology (Biomedical Science), York University, Toronto, ON M3J 1P3, Canada; emilynaucoin@gmail.com; 4Faculty of Arts and Science, Queen’s University, Kingston, ON K7L 3N9, Canada; 21myjc@queensu.ca (M.C.); 18yl210@queensu.ca (Y.L.)

**Keywords:** T1R2/T1R3, G protein-coupled receptors, sweeteners, epithelial-mesenchymal transition, pancreatic cancer, glycosylated receptor, migration, tunneling nanotubes

## Abstract

**Simple Summary:**

Non-nutritive sweeteners (NNS) are widely used by individuals to lower their caloric intake, lose weight, and sustain a healthy diet. The specific mechanistic details of the effects of NNS consumption by individuals on host metabolism and energy homeostasis are unknown. This topic is highly relevant as NNS has been promoted as an option to improve health. However, NNS consumption has been associated with increased risk factors for metabolic syndrome (MetSyn), resulting in diseases like cardiovascular dysfunction, type 2 diabetes mellitus (T2DM), cancer, and metastasis. This report presents a novel molecular mechanism of how NNS and natural sugars binding taste receptors transmogrify glycosylated receptors on cancer cells to induce the epithelial-mesenchymal transition of the metastatic phenotype.

**Abstract:**

Understanding the role of biased taste T1R2/T1R3 G protein-coupled receptors (GPCR) agonists on glycosylated receptor signaling may provide insights into the opposing effects mediated by artificial and natural sweeteners, particularly in cancer and metastasis. Sweetener-taste GPCRs can be activated by several active states involving either biased agonism, functional selectivity, or ligand-directed signaling. However, there are increasing arrays of sweetener ligands with different degrees of allosteric biased modulation that can vary dramatically in binding- and signaling-specific manners. Here, emerging evidence proposes the involvement of taste GPCRs in a biased GPCR signaling crosstalk involving matrix metalloproteinase-9 (MMP-9) and neuraminidase-1 (Neu-1) activating glycosylated receptors by modifying sialic acids. The findings revealed that most natural and artificial sweeteners significantly activate Neu-1 sialidase in a dose-dependent fashion in RAW-Blue and PANC-1 cells. To confirm this biased GPCR signaling crosstalk, BIM-23127 (neuromedin B receptor inhibitor, MMP-9i (specific MMP-9 inhibitor), and oseltamivir phosphate (specific Neu-1 inhibitor) significantly block sweetener agonist-induced Neu-1 sialidase activity. To assess the effect of artificial and natural sweeteners on the key survival pathways critical for pancreatic cancer progression, we analyzed the expression of epithelial-mesenchymal markers, CD24, ADLH-1, E-cadherin, and N-cadherin in PANC-1 cells, and assess the cellular migration invasiveness in a scratch wound closure assay, and the tunneling nanotubes (TNTs) in staging the migratory intercellular communication. The artificial and natural sweeteners induced metastatic phenotype of PANC-1 pancreatic cancer cells to promote migratory intercellular communication and invasion. The sweeteners also induced the downstream NFκB activation using the secretory alkaline phosphatase (SEAP) assay. These findings elucidate a novel taste T1R2/T1R3 GPCR functional selectivity of a signaling platform in which sweeteners activate downstream signaling, contributing to tumorigenesis and metastasis via a proposed NFκB-induced epigenetic reprogramming modeling.

## 1. Introduction

Recent studies have associated NNS consumption with increased risk factors for metabolic syndrome and disease [1]. The specific details and mechanisms of NNS consumption on energy homeostasis and host metabolism remain to be elucidated. However, emerging evidence has revealed that sweet taste receptor signaling dysfunctions may be associated with inflammatory response pathways [2]. When sweet taste-sensing receptors are stimulated by one of four different sweeteners, including sucralose, acesulfame potassium, sodium saccharin, or glycyrrhizin, distinct signaling pathways are activated in pancreatic ß-cells [3]. Patterns of cytoplasmic Ca^2+^ and cAMP-induced changes by these sweeteners were all different from each other. Here, the data support the concept that sweeteners are biased agonists to activate sweet taste-sensing receptors [3].

The gaps in our knowledge regarding how non-nutritive sweetener consumption is implicated in host metabolism via GPCR taste-sensing receptors reinforce our understanding of the mechanistic action of NNS on the body [1,4]. Emerging evidence has also revealed that dysfunctions of sweet taste receptor signaling may be responsible for cognitive impairment in that sweet taste receptor signaling is associated with inflammatory response pathways [2]. The question is what molecular mechanism of biased sweetener GPCR is involved in the inflammatory signaling pathway.

There are reports that bradykinin (BR2) and angiotensin II receptor type I (AT2R) GPCRs exist in a heteromeric GPCR complex with neuromedin-B (NMBR) GPCR and neuraminidase-1 (Neu-1) in naïve and stimulated cells expressing insulin receptor (IRβ) [5] and Toll-like receptor (TLR) [6]. Central to this process is that Neu-1-matrix metalloproteinase-9 (MMP-9) crosstalk, together with NMBR GPCR, is mediated at the ectodomain of these receptors on the cell surface.

This concept of a biased GPCR agonism mediating Neu-1 sialidase and MMP-9 crosstalk to induce transactivation of glycosylated receptor signaling is innovative and fascinating. To this end, this research investigated GPCR heterodimeric T1R2/T1R3 taste-sensing receptors that may exist in an oligomeric receptor complex with NMBR, TLR, and Neu-1 on naïve. It stimulated TLR-expressing cells and other RTK receptors. In support of this concept, studies using knockout mice have revealed that most of the sweet sensation is mediated by the heterodimer of T1R2 and T1R3 [7]. However, a small amount of cellular activity was observed in response to sugars in the T1R3 knockout mice [8]. This observation raised the possibility of an additional mechanism for the detection of sweetener molecules. It seems likely that an alternate sweet-sensing selectivity is mediated by different oligomerizations of related GPCRs on the cell surface. 

Another intriguing observation about the T1R2/T1R3 receptors is the structural diversity of its ligands. This receptor can recognize every sweetener tested, including carbohydrates, amino acids and derivatives, proteins, and synthetic sweeteners [9]. Also, these receptors exhibit stereoselectivity for specific molecules. For example, they respond to D-tryptophan but not L-tryptophan, which is in correlation with the sensory data. It is still a puzzle as to how this single receptor can recognize such an extensive collection of diverse chemical structures. How GPCRs are organized at the cell surface also remains highly contentious. To explain this latter question, Onfroy et al. have proposed that G protein stoichiometry dictates biased agonism through distinct receptor-G protein partitioning [10]. Here, the Gα subunit expression levels may influence the profiling of biased β-agonists as well as antagonists in that they determine both their activity and efficacy by affecting different membrane distributions of the receptor-G protein. Perhaps the most exciting aspect of GPCR biology concerns their stoichiometry, that is, whether they exist as monomers, dimers, or higher-order oligomers on the cell membrane [11].

Furthermore, GPCRs constitute the most abundant class of membrane-bound receptor proteins and, at the basic level, are characterized by seven transmembrane α-helices [12]. These integral proteins are activated by a myriad of ligands, mediating the majority of cellular responses to hormones, neurotransmitters, and environmental stimulants, among others [13]. Activation of the GPCRs can elicit conformational changes in these receptors, which results in the intermediate coupling and activation of guanine triphosphate (GTP)-binding G proteins (guanine nucleotide regulatory proteins, composed of α-, β- and γ-subunits), leading to the exchange of GDP for GTP on the Gα subunit. This interactive process(es) of G proteins results in the dissociation of the GPCR receptor from the G proteins as well as the GαATP subunit from the Gβγ subunit [12,14,15,16,17,18].

GPCR-biased agonists, antagonists, allosteric modulators, or even sweeteners can bind to their GPCRs in different ways, creating unique conformations that differentially modulate signaling through one or more G-proteins or even partner with other receptors for transactivation. The T1R2/T1R3 sweetener receptor has at least six binding sites for the different sweeteners [9]. The initial processing of taste occurs within taste receptor cells, situated in clusters known as taste buds on the tongue [19]. Upon activation by distinct tastants, the taste receptor cells convey information through sensory afferent fibers to specific regions in the brain responsible for taste perception. Four distinct morphological subtypes of taste receptor cells have been discerned [19]. Sweet tastes are detected by G-protein coupled receptors (GPCRs) expressed on type II taste receptor cells [7]. These sweet receptors are composed of two heterodimers of taste-1 receptor member 2 (T1R2) and taste-1 receptor member 3 (T1R3) [7]. These ubiquitous receptors play a significant role in sensing nutrients, monitoring alterations in energy reserves, and initiating metabolic and behavioral responses to uphold energy equilibrium [20,21]. Resultantly, these receptors may play an important role in pathogenesis. The sweet taste receptors can be stimulated by a myriad of chemically distinct compounds, including sugars, artificial sweeteners, sweet proteins, and sweet amino acids [9]. The binding of ligands to the sweet taste receptors activates heterotrimeric G-protein α-gustducin [21]. Subsequent downstream release of ATP results in the activation of adjacent sensory afferent neurons and the relay of signals to brain centers involved in taste receptors [21]. 

Apart from their chemosensory role on the tongue, taste receptors for sweet, umami, and bitter taste are expressed in certain cancers, regulating cellular processes like apoptosis and proliferation [22]. For instance, increased expression of genes encoding bitter receptors (*TAS2R*) in tumors is associated with increased neck and head squamous cell carcinoma overall survival, with apoptosis being activated when bitter bacterial metabolites bind tumor cell bitter GPCR T2R receptors [22]. The binding of bitter agonists to functional TAS2Rs has also been identified to activate apoptosis in the prostate, metastatic breast, acute myeloid leukemia, and pancreatic cancers [22]. Carey and colleagues identified a trend of decreased TAS2R expression amongst malignancies, possibly mediating unregulated proliferation and oncogenesis [22]. However, variability in response to TAS2R activation by bitter ligands has been identified in various cancers, with some having pro-tumor, instead of anti-proliferative, effects in certain malignancies, like submandibular gland cancer cells [22]. Differences amongst positive and negative survival outcomes highlight the diversity of T1R and T2R functions and ligands, with possible synergistic or antagonistic effects in various tumor types [22]. Additionally, genetic changes in cancer cells may mediate taste receptor expression changes and associated survival impacts [22].

The question is whether this biased GPCR sweetener-induced receptor transactivation of the partner receptor tyrosine kinase receptor signaling axes involves Neu-1 sialidase to modify the receptor glycosylation and downstream signaling on cancer cells to induce the epithelial-mesenchymal transition of the metastatic phenotype. 

## 2. Materials and Methods

### 2.1. Cell Lines

Two cell lines were used in this study: PANC-1 (ATCC^®^ CRL-1469™) and RAW-Blue macrophages (InvivoGen, San Diego, CA, USA). We used the Raw-Blue™ cells (Mouse Macrophage Reporter Cell Line, InvivoGen) derived from RAW 264.7 macrophages grown in a culture medium containing Zeocin as the selectable marker [23]. They stably express a secreted embryonic alkaline phosphatase (SEAP) gene, which is inducible by NF-κB and AP-1 transcription factors. Stimulation of RAW-Blue™ cells activates NF-κB and AP-1, leading to SEAP secretion, detectable and measurable using QUANTI-Blue™ (InvivoGen) SEAP in the medium. RAW-Blue™ cells are made to be resistant to Zeocin^®^ (InvivoGen) and G418 (InvivoGen) antibiotics and grown in conditioned media containing Zeocin^®^. The cells were grown in conditioned media 1× DMEM (Dulbecco’s modified eagle medium (Gibco, Rockville, MD, USA), with fetal bovine serum (FBS) at 10% (Hy Clone, Logan, UT, USA) and 5 μg/mL plasmocin (InvivoGen, San Diego, CA, USA). They were maintained at 5% CO_2_ and 37 °C.

### 2.2. Reagents and Inhibitors

In the live cell sialidase assay experiments, 2-(4-methylumbelliferyl)-α-D-N-acetylneuraminic acid (98% pure 4-MUNANA; Biosynth International Inc., Itasca, IL, USA), a sialidase substrate, was used at a concentration of 0.318 mM diluted in tris-buffered saline (TBS). NMBR inhibitor, BIM-23127, was used at 12.5 μg/mL and purchased from Tocris Bioscience, IO Centre Moorend Farm Avenue, Bristol, BS11 0QL, UK. Oseltamivir phosphate (OP) (>99% pure OP, batch No. MBAS20014A, Solara Active Pharma Sciences Ltd., New Mangalore-575011, Karnataka, India), a broad range neuraminidase sialidase inhibitor using predetermined effective dosages, was used at 300 μg/mL MMP-9i inhibitor (Calbiochem-EMD Chemicals Inc., Darmstadt, Germany) is a cell-permeable, which is a potent, selective, and reversible inhibitor at IC_50_ = 5 nM. The MMP-9i also inhibits MMP-13 (IC50 = 113 nM) and MMP-1 (IC_50_ = 1.05 μM) at much higher concentrations. Galardin (GM6001; N-[(2R)-2-(Hydroxamidocarbonylmethyl)-4-methylpentanoyl]-L-tryptophan methyl amide; Calbiochem-EMD Chemicals Inc., Darmstadt, Germany) is a cell-permeable, broad-spectrum hydroxamic acid inhibitor of matrix metalloproteinases (MMPs). TLR4 ligand lipopolysaccharide (LPS) was used at 5 μg/mL from Serratia marcescens and purified by phenol extraction, as per Sigma Aldrich (Millipore Sigma Canada Ltd., Oakville, ON, Canada). Acetylsalicylic acid (ASA, >99% pure, Sigma-Aldrich, Steinheim, Germany) was prepared in dimethyl sulfoxide (DMSO), making aliquots of 5000 mM stock solution and stored at −20 °C. The aspirin concentration has DMSO at 0.5% *v*/*v* in 1× PBS at a pH of 7.

Saccharin sodium salt hydrate (>98 pure, 2,3-Dihydro-3-oxobenzisosulfonazole sodium salt, No. 109185, Sigma-Aldrich, Millipore Sigma Canada Ltd., Oakville, ON L6H 6J8, Canada), advantame (≥97.0%, Sigma-Aldrich), neotame (Sigma-Aldrich), aspartame (Sigma-Aldrich), acesulfame K (Sigma-Aldrich), cyclamate (Sigma-Aldrich), and sucralose (Sigma-Aldrich). Natural sweeteners are stevia (stevia in the raw extract, Cumberland Packing Co. Brooklyn, NY, USA), monk (NatriSweet Organic Monk Fruit Extract, Pure Monk Fruit Sweetener Organic extract with no erythritol), D- (+)-Glucose (Sigma-Aldrich), β-lactose (Sigma-Aldrich), and D- (+)-Galactose (Sigma-Aldrich). All sweeteners were used at a predetermined 200 µg/mL or in a dose-dependent manner. 

### 2.3. Sialidase Assay

PANC-1 and RAW-Blue macrophage cells were cultured and individually grown on a 12 mm circular glass slide in a sterile 24-well tissue culture plate in a conditioned medium for 24 h. Once cells reached approximately 70% confluence, they were serum-starved for 24 h. Media were removed from the wells, and live cells were treated with 4-MUNANA substrate, followed by the treatment of sweetener agonists alone or in combination with an inhibitor at a predetermined concentration. Activated Neu-1 hydrolyzes the 4-MUNANA sialidase substrate, forming free 4-methylumbelliferone (4-MU), which fluoresces at 450 nm (blue color) when excited at 365 nm. Fluorescent images were captured using epi-fluorescent microscopy (Zeiss Imager M2, 20× objective) within 3 min. The sialidase activity was represented by blue fluorescence surrounding the cells’ periphery. The mean fluorescence intensity of 50 different points surrounding the cell was quantified using Image J software (version 1.54 g, Java 1.8.0_345, 64-bit).

### 2.4. NF-κB Dependent Secreted Embryonic Alkaline Phosphatase (SEAP) Assay

Briefly, a cell suspension of 1 × 106 cells/mL in the fresh growth medium was prepared, and 100 μL of RawBlue suspension of cells (~100,000 cells) was added to each well of a Falcon flat-bottom 96-well plate (Becton Dickinson, Mississauga, ON, Canada) [23]. Following varying incubation times, sweetener agonists were added to each well in a dose-dependent manner, either alone or in combination with the specific MMP-9 inhibitor (MMP-9i); oseltamivir phosphate (OP) and BIM-23127 (BIM23) were added to each well 1 h before stimulation with agonists. The plates incubated at 37 °C in a 5% CO_2_ for 18–24 h were followed with QUANTI-Blue™ (InvivoGen) reagent solution as per the manufacturer’s instructions. Briefly, 160 μL of resuspended QUANTI-Blue solution was added to each well of a 96-well flat-bottom plate, adding 40 μL supernatant from the treated RAW-blue cells. Following the plate incubation for 60 min at 37 °C, the SEAP levels were measured using a spectrophotometer (Spectra Max 250, Molecular Devices, Sunnyvale, CA, USA) at 620–655 nm. Each experiment was performed in triplicate. 

### 2.5. Immunocytochemistry

PANC-1 cells plated at a density of 100,000 to 200,000 cells/mL on glass coverslips in 24-well plates were treated with sweeteners at different dosages for 24 h. At the end of the time-point, cells were washed, fixed with 4% PFA for 30 min and blocked for 1 h in 10% FBS + 0.1% Triton X-100 + 1×PBS (0.1% Triton X-100 was omitted from blocking buffer for membrane-only stains to block intracellular non-specific binding). Cells were blocked and washed with 1×PBS, followed by the primary antibody was diluted to 1:250 using a 1% FBS + 1×PBS + 0.1% Triton X-100 overnight at 4 °C. Primary monoclonal IgG antibodies were obtained from Santa Cruz BioTechnology ALDH1A1 (sc-374149), CD24 (sc-19585), E-cadherin and N-cadherin and used at a 1:10 dilution from 200 μg stock. The secondary goat anti-mouse Alexa Fluor 488 antibodies (Santa Cruz Biotechnology, Inc., Dallas, TX 75220, USA) at a concentration of 1:1000 for the immunofluorescence were predetermined standardized protocol. Cells were washed 3× for 10 min with 1×PBS and incubated for 1 h. Secondary antibody controls were included for unspecific staining. Cells were then washed 5× for 15 min with 1×PBS (note: one wash included 0.1% Triton X-100 to permeabilize cells for DAPI). DAPI containing mounting media (Vector Laboratories H-1200-10) was added to slides, and coverslips were inverted onto the mounting media droplet and sealed. Relative fluorescence density readings were quantified using images captured at 20x to ensure a wide field of view was obtained. Two representative images were taken at 20x. Background means, image means, and pixel measurements were obtained from Corel Photo-Paint X8. Red (Alexafluor 594) or green (Alexafluor 488) color channel images quantified. The background mean density represents an unstained section of the image, and the image mean represents the total fluorescence of the image. These measurements were used to quantify the relative fluorescence density using the equation below:Density = (background mean − image mean) × pixels

### 2.6. Tunnelling Nanotubes (TNTs)

PANC-1 cells were cultured and individually grown in a sterile 24-well plate on 12 mm circular glass coverslips containing the conditioned medium for 24 h. Cells were media-starved and incubated with predetermined indicated concentrations of the sweeteners (lactose, neotame, and stevia) in the designated wells for 24 h after reaching 70% confluency. Control wells were incubated with media without FBS. The cells were washed 1x with PBS, treated with Invitrogen CellMask Plasma Membrane Stain (C10045, Thermo Fisher Scientific) and fixed at 4% PFA (300 µL) before being incubated at 4 °C for 24 h. Following this, cells were washed 3 times with PBS-Tween 20 and mounted on microscope slides using 3 μL of VECTASHIELD DAPI fluorescent mounting medium (VECTH1500, MJS BioLynx Inc. 300 Laurier Blvd, K6V 5W1, Brockville, ON, Canada). Slide images were observed using Zeiss M2 epi-fluorescent microscopy (20× magnification, Carl Zeiss Canada Ltd., M3B 2S6 Toronto, ON, Canada), capturing images under the Rhodamine (554 nm) channel. The images were enhanced, and the cell projections were differentiated on ImageJ. The projections were quantified using Corel Photo-Paint. For statistical analysis, we used GraphPad Prism 10. Comparisons between groups from two independent experiments used a one-way analysis of variance (ANOVA) at 95% confidence, followed by Fisher’s uncorrected LSD multiple comparisons post hoc test with 95% confidence. Asterisks denote statistical significance.

### 2.7. Scratch Wound Assay

Cells at 50,000 cells/well were plated in Ibidi cell culture 2 well silicone insert with a defined 500 um cell-free gap on an ibiTreat #1.5 polymer coverslip, tissue culture treated, sterilized µ-Dish 35 mm, and incubated for 24 h at 37 °C in a 5% CO_2_ incubator. Using an Adobe Pro measuring tool, precise and reproducible wound diameters were calculated from captured images at different times created in all wells. Wound width was manually measured at 6–8 points per image (right, middle, left) to obtain an average. Average wound widths were graphed to show wound closure over 8 to 48 h for each respective wound closure. Wound gap closure rate was calculated using the GraphPad Prism software (version 10.2.3.403) to measure the simple linear regression best fit straight line through the data to find the best-fit value of the slope and intercept. The best-fit slope represents the rate in mm/hr of wound closure. 

### 2.8. Statistical Analysis

Data presented as the mean ± the standard error of the mean (SEM) from at least three repeats of each experiment performed in triplicate. Comparisons between groups from two to three independent experiments were made by one-way analysis of variance (ANOVA) at 95% confidence using the uncorrected Fisher’s LSD multiple comparisons test with 95% confidence with asterisks denoting statistical significance.

## 3. Results

### 3.1. Artificial Sweeteners Induce Sialidase Activity in Live RawBlue Macrophage Cells and Is Blocked by Neu-1 Inhibitor Oseltamivir Phosphate (OP), Broad Range MMP Inhibitor Galardin and Neuromedin B Receptor-Specific Antagonist BIM23127

Figure 1A highlights how T1R2/T1R3 sweetener GPCR can facilitate its signaling in forming an oligomeric GPCR complex with partner glycosylated receptors. Here, we performed a live cell fluorometric assay for sialidase activity on a live RawBlue macrophage cell line, each treated with optimal doses of high-intensity sweeteners (aspartame, acesulfame-potassium (Ace-K), neotame, advantame). We also performed a fluorometric assay on the cells treated with an optimum dose of sweetener with oseltamivir phosphate (OP) and acetylsalicylic acid (ASA), galardin and BIM23127 and MMP-9i at optimal inhibitory doses, as previously reported [6]. Here, all inhibitors had a significant effect on inhibiting sialidase activity on acesulfame-potassium, neotame and aspartame sweetener-stimulated live RawBlue macrophage cells at indicated concentrations of the agonists (Figure 1B,C).

### 3.2. Natural Sweeteners Glucose, Stevia and Monk Induce Sialidase Activity in Live RawBlue Macrophage Cells and Are Inhibited by Neu-1 Inhibitor Oseltamivir Phosphate (OP), MMP-9i Inhibitor and Neuromedin B Receptor-Specific Antagonist BIM23127

We questioned whether natural sweeteners like glucose, stevia and monk induce sialidase activity in live RawBlue macrophage cells using a similar oligomeric functional selectivity of T1R2/T1R3 GPCR-biased heteromers. To test this hypothesis, Neu-1 inhibitor OP, MMP-9i inhibitor and neuromedin B receptor-specific antagonist BIM23127 (BM-23) should inhibit natural sweetener-induced Neu-1 sialidase activity.

As shown in Figure 2A, stevia and monk sweeteners induced a markedly significant sialidase activity compared to the control background, while glucose induced a minimally significant sialidase activity. Furthermore, Neu-1-specific inhibitor OP, MMP-9i and BIM23127 significantly inhibited stevia and monk-induced sialidase activity (Figure 2B,C). These data support the concept that natural sweeteners promote oligomeric functional selectivity of T1R2/T1R3 GPCR-biased heteromers.

### 3.3. Artificial and Natural Sweeteners Induce Sialidase Activity in Live Pancreatic PANC-1 Cancer Cells

Figure 3A depicts Neu-1 sialidase in complex with MMP-9 and NMBR regulates receptor tyrosine kinases (RTKs) expressed on cancer cells. Here, we asked if the oligomeric functional selectivity of T1R2/T1R3 GPCR-biased natural and artificial sweeteners can induce Neu-1 sialidase activity compared to the control background in pancreatic PANC-1 cancer cells. As depicted in Figure 3B, natural glucose, stevia and monk, and the artificial sweeteners advantame, neotame, aspartame, acesulfame K, cyclamate, sucralose, and saccharin at predetermined dosage bind T1R2/T1R3 taste GPCR to form an oligomeric functional selectivity of biased heteromers with NMBR (Figure 3A) to induce a significant sialidase activity in PANC-1 cells (Figure 3B). 

### 3.4. Artificial and Natural Sweeteners Induce NFκB-Dependent SEAP Activity in Live RawBlue Macrophage Cells

We asked if artificial and natural sweeteners would directly induce NF-κB in the absence of any TLR-specific ligand in live RawBlue macrophage cells. The data in Figure 4 are consistent with this concept. Here, saccharin, aspartame, acesulfame-potassium (Ace-K), cyclamate, and galactose bind to T1R2/T1R3 GPCR which heterodimerized with neuromedin B receptor (NMBR) tethered to TLR receptors (Figure 4A) and induced NFκB-dependent secretory alkaline phosphatase (SEAP) activity in live RawBlue macrophage cells compared to the positive control LPS in a dose-dependent manner (Figure 4B). It is interesting to note that the sweeteners have similar activating NFκB responses as the TLR4 ligand LPS. Notably, galactose-induced NFκB-dependent SEAP required 1000x dosage than the artificial sweeteners.

### 3.5. Artificial Sweeteners Saccharin, Acesulfame and Advantame but Not Glucose Induce the Acquisition of Stem-like and Invasive Epithelial-Mesenchymal Transition (EMT) Phenotypic Markers

Cell growth markers of chemoresistance and cancer stem-like (CSC) properties include alcohol dehydrogenase-1 family member A1 (ALDH1), CD24, and CD44. High CD44 expression and low expression of CD24 are some of the marker characteristics. Aldehyde dehydrogenase-1 (ALDH1) is also used to characterize cancer cell stemness. The enzyme ALDH1 is involved in the retinol oxidation to retinoic acid, which is essential for the early differentiation of stem cells. As shown in Figure 5, PANC-1 cells treated with saccharin and acesulfame revealed a significant decrease in the expression of CD24 and a significant increase in ALDH1 compared to the untreated group. At the same time, advantame and glucose had no significant effects. Saccharin, acesulfame and advantame increased N-cadherin expression.

### 3.6. Half Maximal Effective Concentration (EC50) of Artificial Sweeteners Saccharin, Acesulfame and Advantame to Induce an Invasive Epithelial-Mesenchymal Transition (EMT) N-Cadherin Phenotypic Marker

The data depicted in Figure 5D revealed that the artificial sweeteners saccharin, acesulfame, and advantame induced a significant expression of N-cadherin in PANC-1 pancreatic cancer cells. Here, we investigated the potency of the sweetener by grading the dose-response curve and calculating the half-maximal effective concentration (EC50) for each of the sweeteners to induce N-cadherin expression. The data in Figure 6 reveal differential half-maximal effective concentration values for saccharin (EC50 = 6.446 mg/mL, 35.2 mM), acesulfame (EC50 = 83.23 mg/mL, 413.5 mM), and advantame (EC50 = 9.899 mg/mL, 72.7 mM) to express N-cadherin for PANC-1 invasion and metastasis.

### 3.7. Half Maximal Effective Concentration (EC50) of Natural Sweeteners Lactose, Monk, and Stevia to Induce an Invasive Epithelial-Mesenchymal Transition (EMT) N-Cadherin Phenotypic Marker

In addition, we investigated the potency of the natural sweeteners lactose, monk, and stevia by grading the dose-response curve and calculating the half-maximal effective concentration (EC50) for each of the sweeteners to induce N-cadherin expression. The data in Figure 7 reveal differential half-maximal effective concentration values for lactose (EC50 = 1.321 mg/mL, 3.86 mM), monk (EC50 = 1.294 mg/mL, 1.005 mM), and stevia (EC50 = 19.5 pg/mL, 0.0242 mM) to express N-cadherin for PANC-1 invasion and metastasis.

It is interesting to note that lactose is a disaccharide of galactose and glucose subunits, which form a β-1→4 glycosidic linkage. Lactose’s systematic name is β-D-galactopyranosyl-(1→4)-D-glucose. The glucose can be in either the α-pyranose form or the β-pyranose form, whereas the galactose can only have the β-pyranose form (Figure 8A). The D- and L-glucose can also activate T1R2 and T1R3 [9]. Interestingly, only lactose induced a markedly different N-cadherin expression in PANC-1 cells compared to no or minimal N-cadherin expression using glucose and galactose (Figure 8B). 

### 3.8. Artificial and Natural Sweeteners Enhance the Migratory and Invasion Potentials of Pancreatic PANC-1 Cancer Cells in a Scratch Wound Assay

It is noted that both artificial and natural sweeteners induce an invasive EMT N-cadherin expression following 24-h exposure in a dose-dependent manner, as depicted in Figure 6 and Figure 7. These observations suggest that the effects of these sweeteners may also influence the migratory potential of cancer cells. To confirm this concept, we investigated the migratory invasiveness of PANC-1 cancer cells treated with stevia, lactose, advantame and neotame in a scratch wound assay. The data in Figure 9 depict the migratory rate of these sweeteners from the scratch wound area over 8 to 24 h. The untreated control cells showed near-complete wound closure after 24 h with a wound closure rate of 2.16 ± 0.062 mm/hr. In contrast, the wound closure for stevia is 9.93 ± 0.61 mm/hr, lactose 6.36 ± 0.63 mm/hr, advantame 9.66 ± 0.5 mm/hr, and neotame 6.36 ± 0.57 mm/hr occurring within 7 to 12 h. These data suggest that artificial and natural sweeteners influence the migration and invasiveness of PANC-1 cancer cells by potentially altering epithelial−mesenchymal transition (EMT) for metastasis.

### 3.9. Artificial and Natural Sweeteners Induce Tunneling Nanotubes (TNTs), Staging the Migratory Intercellular Communication in the Tumor Microenvironment

Cancer cells have developed the unique TNT tool to overcome the challenge of signal transport through rigorous tumor structures within the tumor microenvironment. It can provide spatial restriction and specificity of communications for invasion.

To this end, we investigated the ability of lactose, stevia, and neotame at 100 µg/mL each to form PANC-1 TNTs after 24 h in culture. The data depicted in Figure 10 support the concept of their ability to initiate their communications for invasion induced with lactose, neotame and stevia. It is noteworthy that there are changes in the shape of the treated cell compared to the untreated control. The control group is rounder, while the treated cells are more jagged in shape. When cancer cells metastasize, they morph, becoming missile-like in shape so that they can penetrate other tissues throughout the body.

## 4. Discussion

This study investigated the oligomeric functional selectivity of T1R2/T1R3 GPCR-biased artificial and natural sweeteners on transactivating glycosylating receptors to induce epithelial-mesenchymal transition of the metastatic phenotype. The emerging evidence to support this GPCR selective signaling paradigm has revealed that dysfunctional biased functional selectivity of sweet taste receptor signaling may be associated with one or more different sweeteners, including sucralose, acesulfame potassium, sodium saccharin, or glycyrrhizin where there are distinct activated signaling pathways in pancreatic ß-cells [3]. The changing patterns in cytoplasmic Ca^2+^ and cAMP induced by these sweeteners in pancreatic ß-cells were all different from each other. The findings supported the concept that sweeteners are biased agonists to activate sweet taste-sensing receptors [3]. Other studies using knockout mice have revealed that most of the sweet sensation is mediated by the heterodimerization of T1R2 and T1R3 [7]. However, a small amount of the cellular activity that was observed in response to the sugars was observed in the T1R3 knockout mice [8]. This observation raised the possibility of an alternate sweet-sensing activity that is mediated by different oligomerizations of related GPCRs on the cell surface. In support of this concept, we have also identified a novel oligomeric functional signaling platform of which T1R2/T1R3 taste-sensing receptors form a multimeric receptor complex with neuromedin NMBR, Neu-1, and RTK or TLR on naïve pancreatic cancer cells and macrophage cells, respectively (Figure 10, graphical model). Here, the dimeric taste TIR2/TIR3 GPCR receptors can regulate the interaction and signaling mechanism(s) between these molecules on the cell surface. This molecular model is proposed to uncover a biased TIR2/TIR3 GPCR agonist-induced receptor transactivation signaling axis, mediated by Neu-1 activity, the modification of receptor glycosylation and downstream signaling for cellular activity.

In support of this concept and study model of T1R2/T1R3 sweetener GPCR, Abdulkhalek et al. [6] have demonstrated that GPCR agonists bradykinin, bombesin, cholesterol, lysophosphatidic acid (LPA), angiotensin-1 and -2, but not thrombin-induced Neu-1 sialidase activity in live macrophage cell lines but not from Neu-1-deficient mice. Also, Haxho et al. [5] reported that GPCR angiotensin II (type 1) receptor (AngIIR1) co-localizes with Neu-1 and co-immunoprecipitated with both NMBR and insulin receptor IRβ in naïve, unstimulated HTC-IR. 

Kojima et al. [3] reported that sweeteners are biased agonists to activate sweet taste-sensing receptors. Here, we have demonstrated that artificial sweeteners like acesulfame-potassium (Ace), advantame (Adv), neotame (Neo), and aspartame revealed an oligomeric functional selectivity of T1R2/T1R3 GPCR-biased heteromers with NMBR to induce the Neu-1 sialidase activity of TLR activation signaling axis. Here, the data depicted in Figure 1B clearly demonstrated that the artificial sweeteners aspartame, acesulfame-potassium (Ace-K), neotame, and advantame each significantly induced sialidase activity in live RAW-Blue macrophages in vitro. In addition, natural sweeteners stevia, monk and glucose also induced a significant sialidase activity compared to the control background. Furthermore, oseltamivir phosphate (OP), a specific inhibitor of Neu-1, inhibitor of MMP-9 (MMP-9i) and the specific inhibitor of NMBR BIM23127, significantly inhibited sialidase activity with the artificial sweeteners (Figure 1C,D) as well as the natural sweeteners (Figure 2B,C). These data support our proposed signaling paradigm, where artificial and natural sweeteners binding T1R2/T1R3 taste-sensing receptors induce NMBR-MMP-1 crosstalk to induce Neu-1 sialidase activity. These findings support the concept that sweeteners promote oligomeric functional selectivity of T1R2/T1R3 GPCR-biased heteromers in partnership with glycosylated receptors on the cell surface.

Another intriguing observation about the T1R2/T1R3 receptor is the structural diversity of its ligands. This receptor can recognize every sweetener tested, also including carbohydrates, amino acids and derivatives, proteins, and synthetic sweeteners [9] (see Figure 3A). Also, T1R2/T1R3 exhibits stereo selectivity for different molecules and can recognize an extensive collection of diverse chemical structures. The question is whether biased sweeteners can bind to T1R2/T1R3 GPCRs in different ways, creating unique conformations that differentially modulate signaling through one or more G proteins or form a partnership with RTK or TLR cell surface receptors for transactivation. Does this biased GPCR sweetener-induced partnership of receptor transactivation signaling axis induce Neu-1 sialidase to modify receptor glycosylation for downstream cellular signaling?

Since Neu-1 activity is associated with GPCR signaling and MMP-9 activation in live TLR-expressing macrophage cells [24], we asked if T1R2/T1R3 GPCR sweeteners would directly induce NF-κB in the absence of any TLR-specific ligand. The data in Figure 4B are consistent with this hypothesis. Here, saccharin, aspartame, acesulfame-potassium (Ace-K), cyclamate and galactose binding to T1R2/T1R3 GPCR heterodimerized with neuromedin B receptor (NMBR) tethered to TLR receptors and induced NFκB-dependent secretory alkaline phosphatase (SEAP) activity in live RawBlue macrophage cells compared to the positive control LPS in a dose-dependent manner. Notably, since the data depicted in Figure 1 and Figure 2 showed a link between sweetener T1R2/T1R3 receptor and Neu-1 sialidase activity, it has been reported that Neu-1 can influence the expression of epithelial-mesenchymal transition (EMT) markers such as E-and N-cadherins [25]. Interestingly, Huber et al. [26] reported that NF-κB activation is also essential for EMT and metastasis in breast cancer progression. 

Mammalian Neu-1 has been reported to regulate the activation of several receptor tyrosine kinases (RTKs) [27], all of which are upregulated in cancer and their downstream signaling pathways [28]. This RTK signaling platform plays critical roles in ligand-induced activation of tumor progression, critical compensatory signaling mechanisms, EMT programs, cancer stem cells (CSC), and metastases in human pancreatic cancer [29]. All of these receptors are activated in cancer cells. When the growth factor ligand binds to its RTK receptor, the receptor undergoes a conformational change, which results in the activation of MMP-9 via Gi subunit signaling to remove the elastin binding protein (EBP). The removal of EBP activates Neu-1 in complex with the protective protein cathepsin A (PPCA) [30]. Activated Neu-1 in complex with the receptor at the ectodomain hydrolyzes terminal α-2,3-sialyl residues to remove steric hindrance for receptor dimerization and downstream signaling pathways [27]. OP targeting Neu-1 has also been reported to downregulate several EGFR-mediated pathways, such as the JAK/STAT, PI3K/Akt, and MAPK pathways, involved in cancer cell proliferation, metastasis, and tumor vascularization [31]. 

In support of this hypothesis, the data depicted in Figure 5 show that saccharin and acesulfame showed a significant marked decrease in the expression of CD24 and a significant increase in ALDH1 compared to the untreated group. At the same time, advantame and glucose had no significant effects. Saccharin, acesulfame and advantame, on the other hand, increased N-cadherin expression. The data in Figure 6 reveal differential half maximal effective concentration values for saccharin (EC50 = 35.2 mM), acesulfame (EC50 = 413.5 mM), and advantame (EC50 = 72.7 mM) to express N-cadherin for PANC-1 invasion and metastasis. Interestingly, the potency of the natural sweeteners’ lactose, monk and stevia in Figure 7 revealed a much lower half maximal effective concentration values for lactose (EC50 = 3.86 mM), monk (EC50 = 1.0 mM), and stevia (EC50 = 0.0242 mM) to express N-cadherin for PANC-1 invasion and metastasis than the artificial sweeteners. Surprisingly, stevia was 100-fold more potent than lactose and monk in inducing N-cadherin in PANC-1 cells.

Since lactose is a disaccharide sugar composed of galactose and glucose subunits, we were interested in investigating lactose, galactose, and glucose to induce N-cadherin expression in PANC-1 cells. The data depicted in Figure 8 reveal that lactose markedly induced N-cadherin in PANC-1 cells compared to a negligible N-cadherin induction with glucose and galactose. Interestingly, biased lactose may bind to T1R2/T1R3 GPCRs in a different steric selective way, creating a unique conformation that differentially modulates signaling through a partnership with RTK cell surface receptors for cellular transactivation to induce NFκB activation and N-cadherin expression as depicted in Figure 9. 

If this biased functional selectivity of sweeteners can modulate signaling through a partnership with RTK cell surface receptors for cellular transactivation, we investigated whether artificial and natural sweeteners would induce tunneling nanotubes (TNTs) in staging the intercellular cancer cell communications to enhance the migratory and invasion potentials of pancreatic PANC-1 cancer cells in a scratch wound assay. Mounting evidence suggests that intercellular communication by TNTs may contribute to tumor survival and progression [32]. TNTs have also been associated with cancer cell invasion. Notably, TNTs have been proposed to be critically involved in tumor initiation, growth, progression, metastasis, and chemotherapy resistance [32]. Here, lactose, stevia and neotame at 100 ug/mL for 24 h significantly increased the TNTs in staging the communications of the cancer cells for migratory properties of PANC-1 cells (Figure 10). Indeed, the migratory rate of stevia was 9.93 ± 0.61 mm/hr, lactose 6.36 ± 0.63 mm/hr advantame 9.66 ± 0.5 mm/hr, and neotame 6.36 ± 0.57 mm/hr occurring within 7 hr for wound closure compared to untreated control of 2.16 ± 0.062 mm/hr over 24 hr (Figure 9). These data suggest that artificial and natural sweeteners trigger the migratory communications of the cancer cells to induce epithelial-mesenchymal transition (EMT) for metastasis.

Recent reports on the metabolism of lactose have also indicated that a daily intake of 10g of lactose increased the risk of ovarian cancer by 13% [33]. In addition, a case-control study found that high lactose intake is associated with an increased risk of pancreatic cancer [34]. Lactose is known to regulate insulin secretion [35]. Whey protein consumption, which contains lactose and galactooligosaccharide, is linked to NF-κB signaling [36]. A recent study found that 2′-fucosyllactose and 6′-sialyllactose, derivatives of lactose, inhibited TLR-4 activation, although lactose itself did not [37]. In addition, sialyl (α-2,3) lactose has been reported to interact with the TLR4 receptor, which is linked to intestinal inflammation, and the sialyl (α-2,3) lactose inhibited the MAPK/ERK1/2 downstream signaling in acinar pancreatic cells [1]. 

Interestingly, stevia, compared to lactose and monk, had the highest potency (EC50 = 0.0242 mM) to express N-cadherin for PANC-1 invasion and metastasis (Figure 7F). Other researchers have reported that stevia may have anti-hypertensive, anti-obesity, anti-diabetic, antioxidant, anti-cancer, anti-inflammatory, and antimicrobial effects, along with improving kidney function [38]. Other reports have shown that stevia extracts may have anti-cancer effects; for example, steviol glycosides inhibited human gastrointestinal cancer cell proliferation, inhibited Epstein Barr virus early induction to inhibit tumor progression, and had more significant cytotoxicity for cancer cells compared to normal cells [38]. Additionally, stevioside glycoside has been shown to decrease colon cancer cell viability, inhibit DNA synthesis, and induce apoptotic cancer cell death through mitochondrial apoptosis [38]. Specifically, steviol upregulated the Bax/Bcl-2 ratio via increasing p21 and p53 protein expression and decreasing cyclin D; consequently, Bax protein initiates mitochondrial apoptosis and releases cytochrome c that activates caspases that cleave enzymes responsible for repairing DNA and upholding genome integrity [38].

Furthermore, treating RAW 264.7 macrophage cells with ethyl acetate extracted from stevia leaves significantly inhibited NFκB-mediated gene expression induced by bacterial lipopolysaccharides, consequently decreasing interleukin-6 and monocyte chemoattractant protein-1 concentrations [38]. Notably, the chemical composition of Stevia varies in dry and fresh leaves, including the processing or extraction methods, as well as the geographical region of growth [38]. Therefore, the composition of stevia extracts would have varied effects on the host’s metabolic homeostasis.

Similar findings to stevia extract compositions and their effects on host metabolism, monk fruit extract sweetness is due to possessing the mogroside glycoside of cucurbitane derivatives. Mogroside V is about 250 times sweeter than sucrose and is non-caloric [39]. Also, the in vitro and in vivo studies have suggested mogrosides possess antioxidant and anti-inflammatory effects, as well as the extracts mogroside V and 11-oxo-mogroside V possessed inhibitory effects on mouse skin carcinogenesis models [39]. From the triterpenoid glycoside extract from monk fruit, Liu et al. [40] found that mogroside IVe inhibited colorectal cancer HT29 cells and throat cancer Help-2 cell proliferation in a dose-dependent manner in culture and xenografted mice. Additionally, mogroside IVe upregulated tumor suppressor p53 while downregulating MMP-9 and phosphorylated extracellular signal-regulated kinases (ERK)½ [40]. Importantly, future research on chronic toxicity and carcinogenicity testing requires consideration of the processing of natural sweetener’s extraction composition is highly warranted [41].

## 5. Conclusions

NNS continues to be used as a staple in the Western diet. However, its impacts on health-related safety are poorly studied and understood. Here, we have uncovered a new signaling paradigm for these receptors and cell responses, which could represent a common metabolic target with pro-inflammatory responses. The relationship between sweetener-induced TIR2/TIR3 receptor signaling to activate RTK and TLR receptors could explain the signaling pathways that result in insulin resistance, diabetes, cancer, and metastasis. In turn, the mechanisms leading to diet-induced epigenetic rewiring-inducing metabolic changes contributing to diabetes, metabolic syndrome, tumorigenesis, and metastatic disease can be assessed in future studies (Figure 11). Shigemura et al. [42] demonstrated that the expression of AngII GPCR receptors interacts with T1R3 GPCR in taste tissues and gustatory nerve and behavioral responses to taste stimuli after administration of AngII in mice, depicted in Figure 11. This concept elegantly substantiates the conceptual framework proposing the activation of the biased T1R2/T1R3 GPCR-Neu-1-MMP-1 signaling platform by a select group of sweeteners activating the NF-κB pathway and subsequent epigenetic remodeling, implicated in enhanced EMT markers and metastasis [43]. Although the role of the T1R2/T1R3 GPCR-Neu-1-MMP-9 signaling platform and the associated activation of NF-κB has yet to be studied concerning cancer epigenetics and metastasis, NF-κB presents a significant potential as a mediator of epigenetic effects induced by sweeteners.

## Figures and Tables

**Figure 1 nutrients-16-01840-f001:**
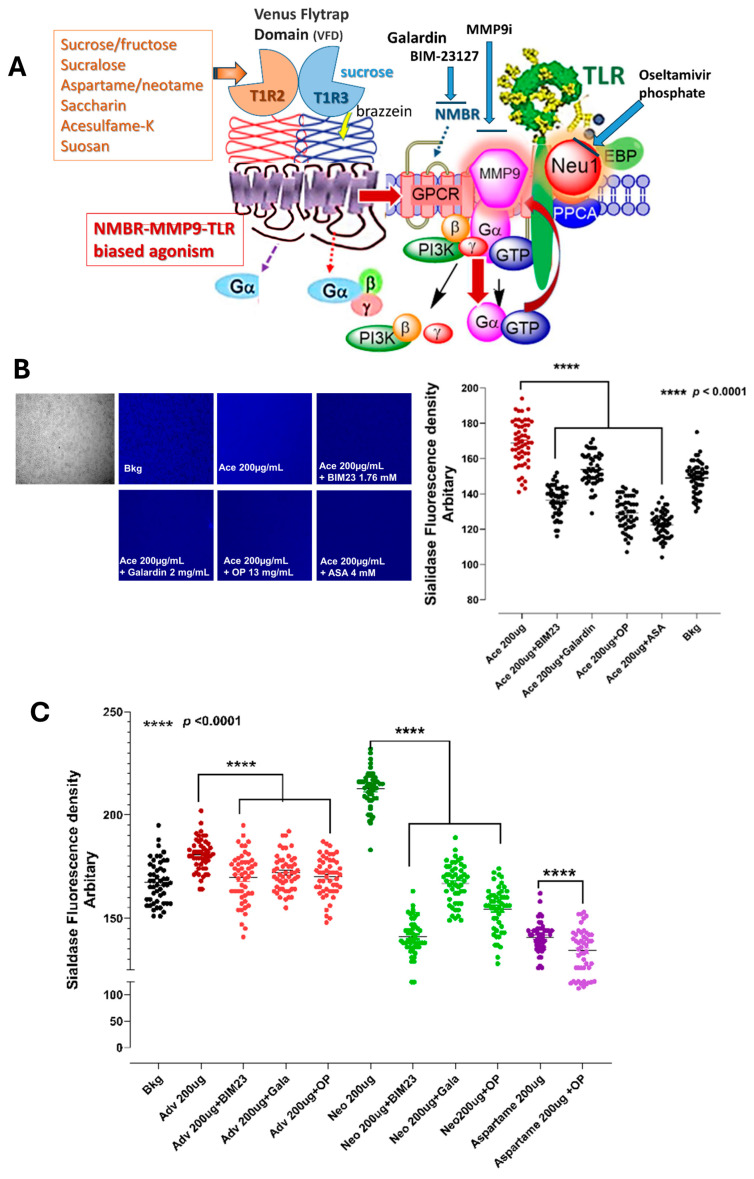
(**A**) Molecular signaling platform on the cell surface reveals an oligomeric selectivity of T1R2/T1R3 GPCR-biased functional heteromers with NMBR to induce the TLR activation signaling axis. Citation: Taken in part from Abdulkhalek et al. [6] Cellular signalling 2012, 24, 2035–2042. Publisher and licensee Elsevier. This is an Open Access article which permits unrestricted non-commercial use, provided the original work is properly cited. (**B**,**C**) Sialidase activity is associated with artificial sweeteners treatment of live RAW-blue macrophage cells. The phase contrast image of the cells represents the cell numbers used in the sialidase assay.

**Figure 2 nutrients-16-01840-f002:**
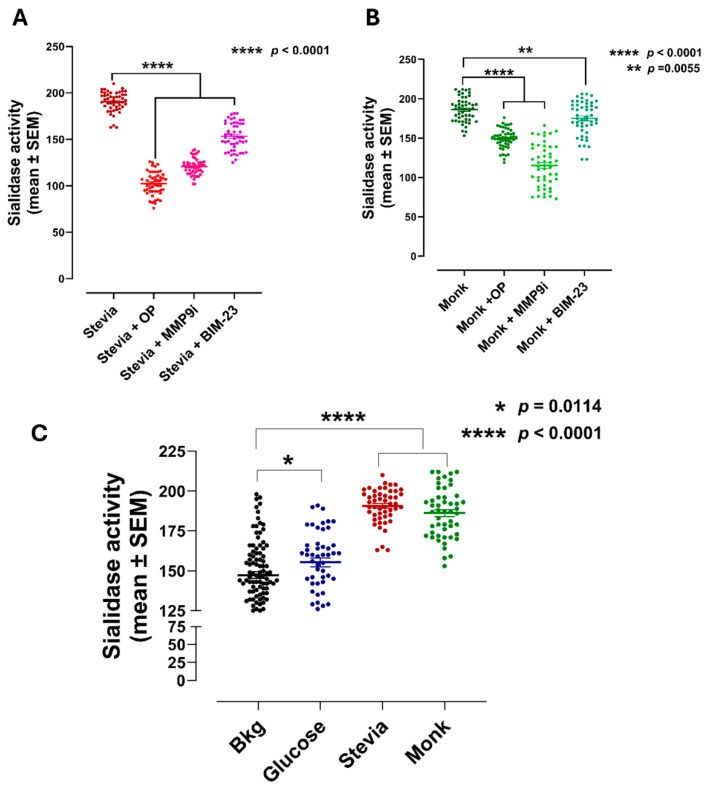
(**A**–**C**) Sialidase activity is associated with natural sweeteners glucose, stevia and monk using live RAW-blue macrophage cells.

**Figure 3 nutrients-16-01840-f003:**
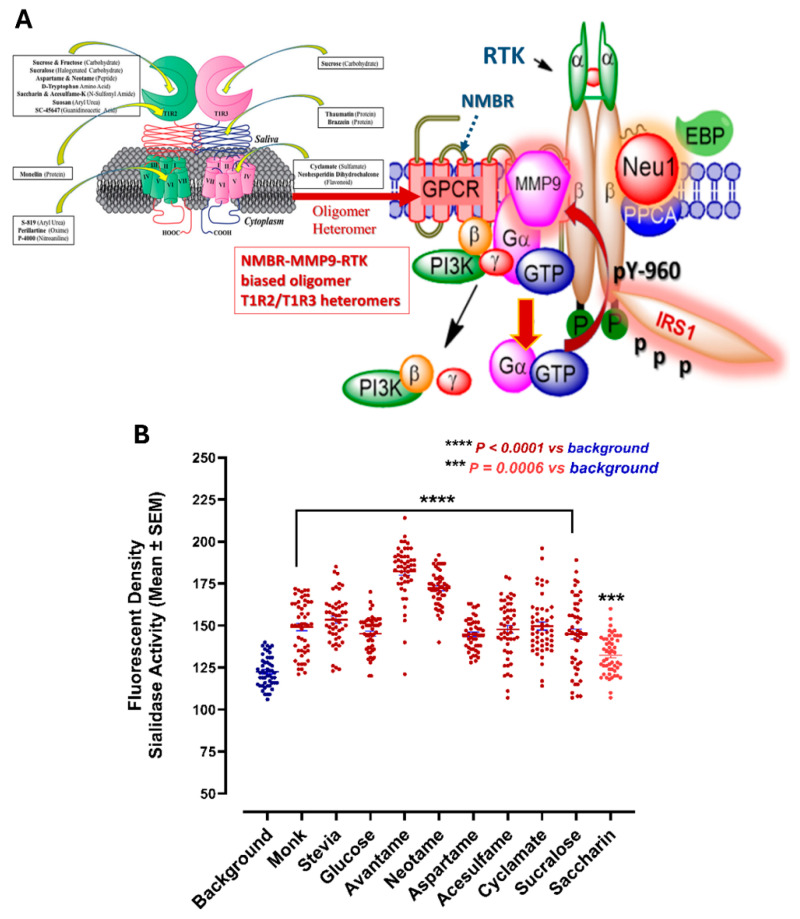
(**A**) Molecular signaling platform reveals an oligomeric selectivity of T1R2/T1R3 GPCR-biased functional heteromers with NMBR to induce the RTK activation signaling axis. This signaling platform potentiates MMP-9 and Neu-1 crosstalk on the cell surface, which is essential for activating RTK. Citation: Taken in part from DuBois [12] Physiology & Behavior 2016, 164, 453–463, Publisher and licensee Elsevier. and Liauchonak et al. [4] Nutrients 2019, 11, 644, Publisher and licensee MDPI. This is an Open Access article which permits unrestricted non-commercial use, provided the original work is properly cited. (**B**) Sialidase activity is associated with natural and artificial sweeteners treatment of live pancreatic PANC-1 cancer cells.

**Figure 4 nutrients-16-01840-f004:**
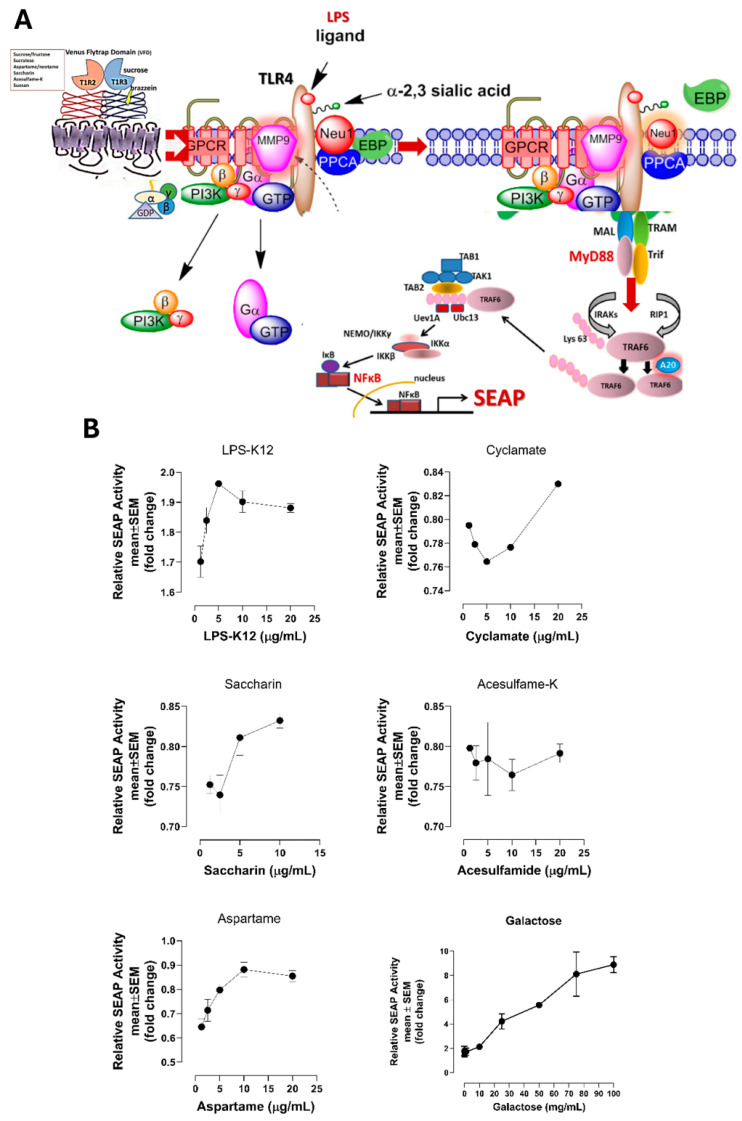
(**A**) Sweetener T1R2/T1R3 GPCR agonists activate NMBR GPCR tethered to TLR4 and MMP-9 to induce Neu-1 sialidase in macrophage cells. Citation: Taken in part from Abdulkhalek et al. [6] Cellular signalling 2012, 24, 2035–2042. Publisher and licensee Elsevier. This is an Open Access article which permits unrestricted non-commercial use, provided the original work is properly cited. (**B**) Artificial and natural sweeteners induce NFκB-Dependent SEAP activity in live RawBlue macrophage cells.

**Figure 5 nutrients-16-01840-f005:**
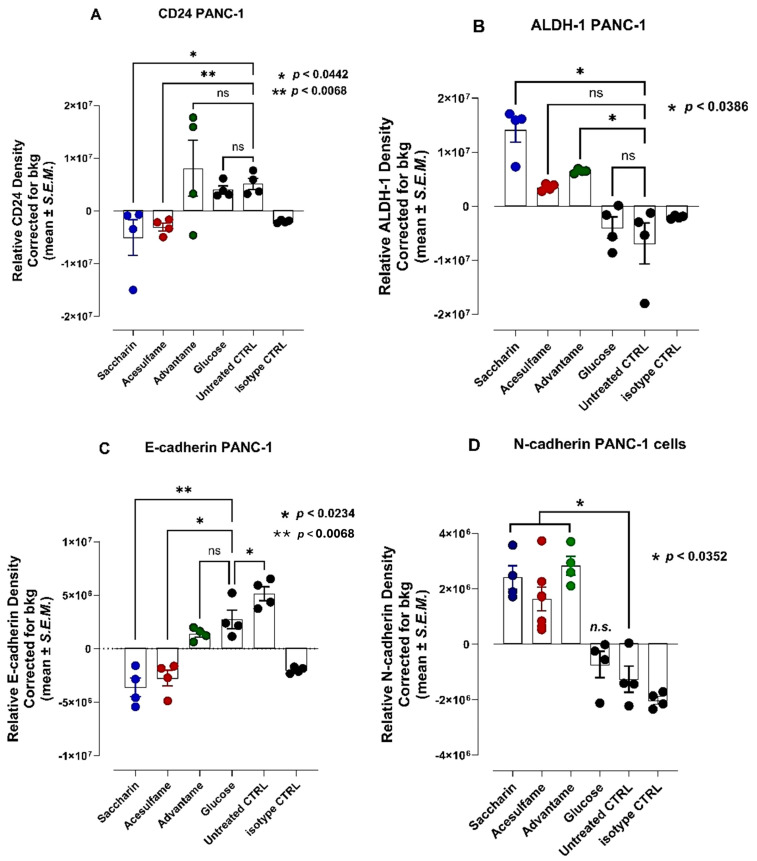
Analyses of pancreatic PANC-1 cancer cells for (**A**) CD24 and (**B**) ALDH1A1 stem-like markers and (**C**) E-and (**D**) N-cadherin EMT markers following 24 h exposure to saccharin, acesulfame, advantame and glucose using immunocytochemistry analyses. ns = non-significant.

**Figure 6 nutrients-16-01840-f006:**
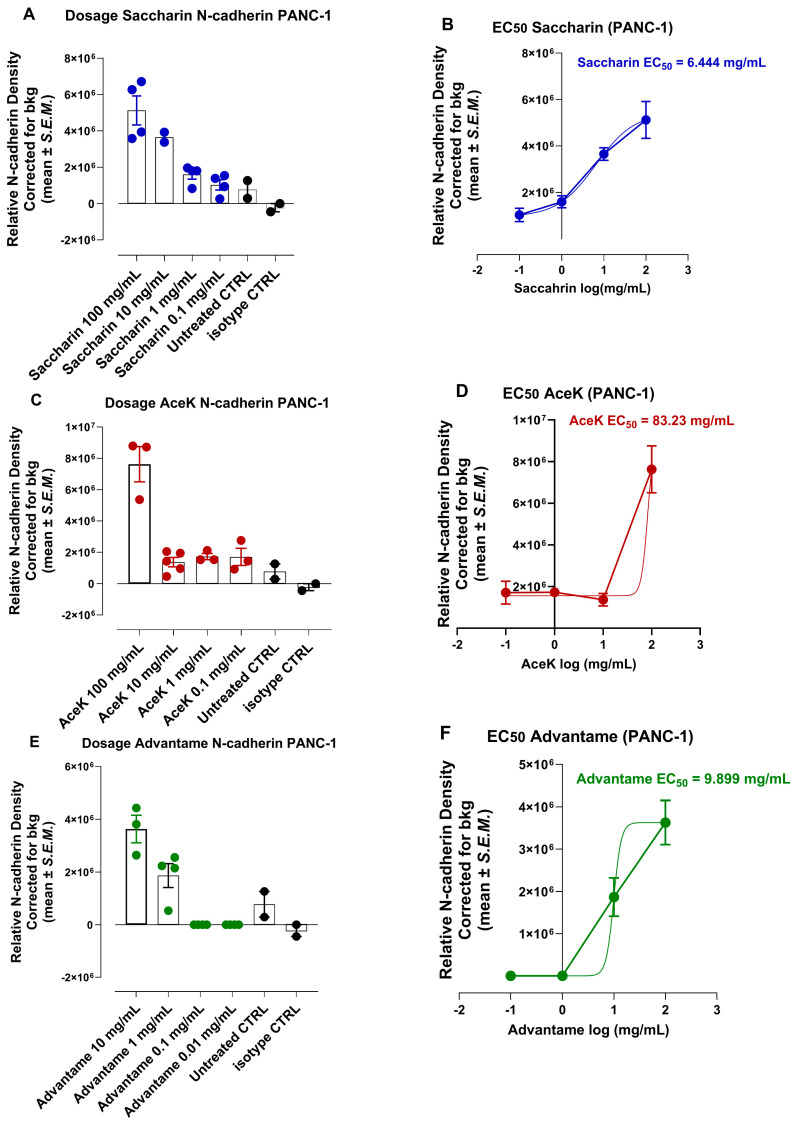
Half maximal effective concentration (EC50) of artificial sweeteners (**A**,**B**) saccharin, (**C**,**D**) acesulfame K and (**E**,**F**) advantame to induce an invasive epithelial-mesenchymal transition (EMT) N-cadherin expression following 22 h exposure using immunocytochemistry analyses.

**Figure 7 nutrients-16-01840-f007:**
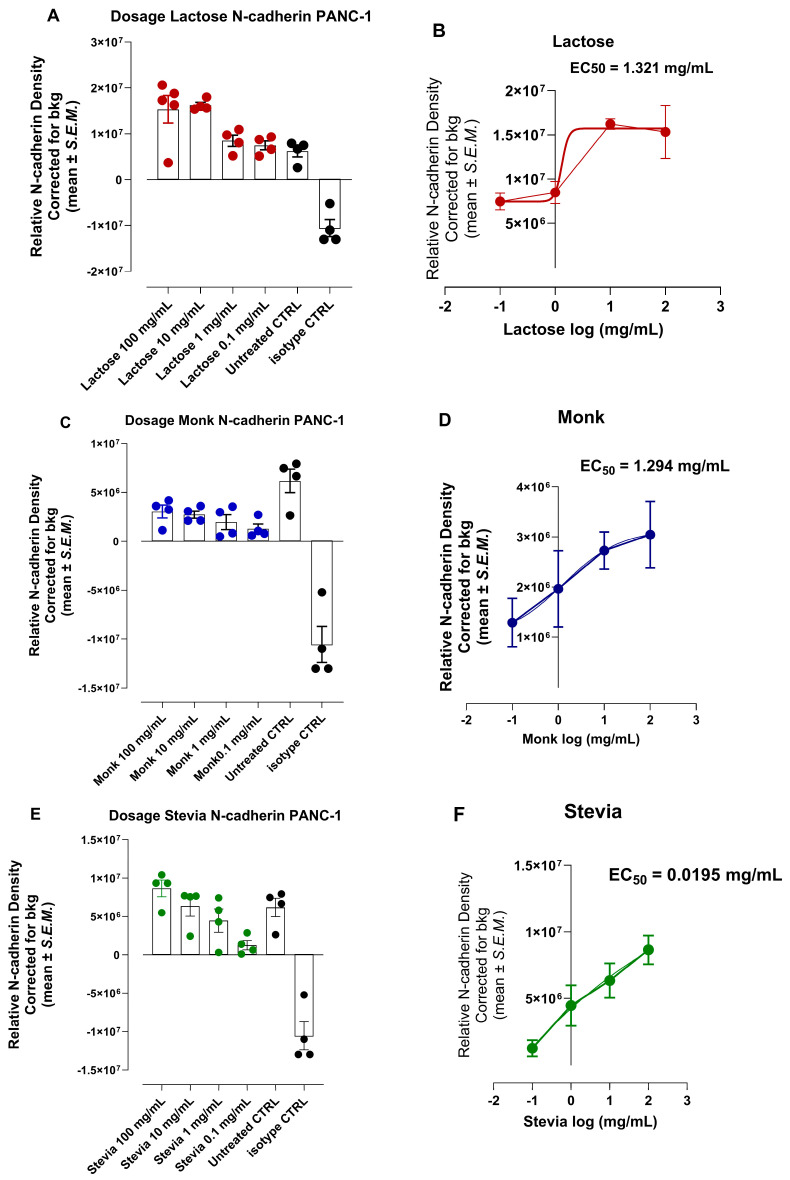
Half maximal effective concentration (EC50) of natural sweeteners (**A**,**B**) lactose, (**C**,**D**) monk and (**E**,**F**) stevia to induce an invasive epithelial-mesenchymal transition (EMT) N-cadherin expression following 24 h exposure using immunocytochemistry analyses.

**Figure 8 nutrients-16-01840-f008:**
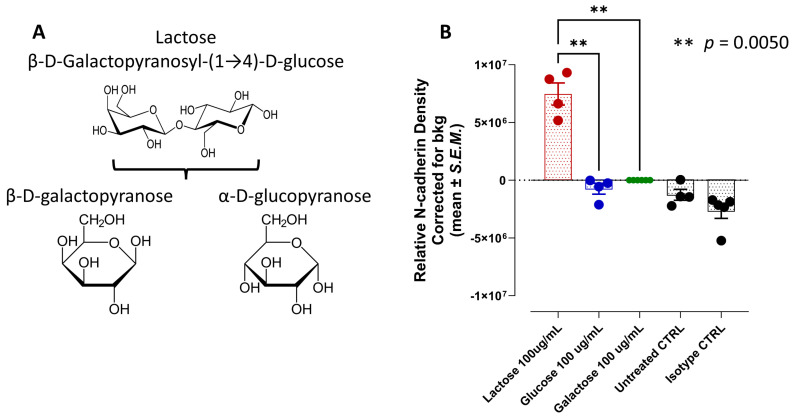
(**A**) Structures of lactose, galactose, and glucose. (**B**) Natural sweeteners lactose, and not glucose and galactose, induce an invasive epithelial-mesenchymal transition (EMT) N-cadherin expression following 24-h exposure using immunocytochemistry analyses.

**Figure 9 nutrients-16-01840-f009:**
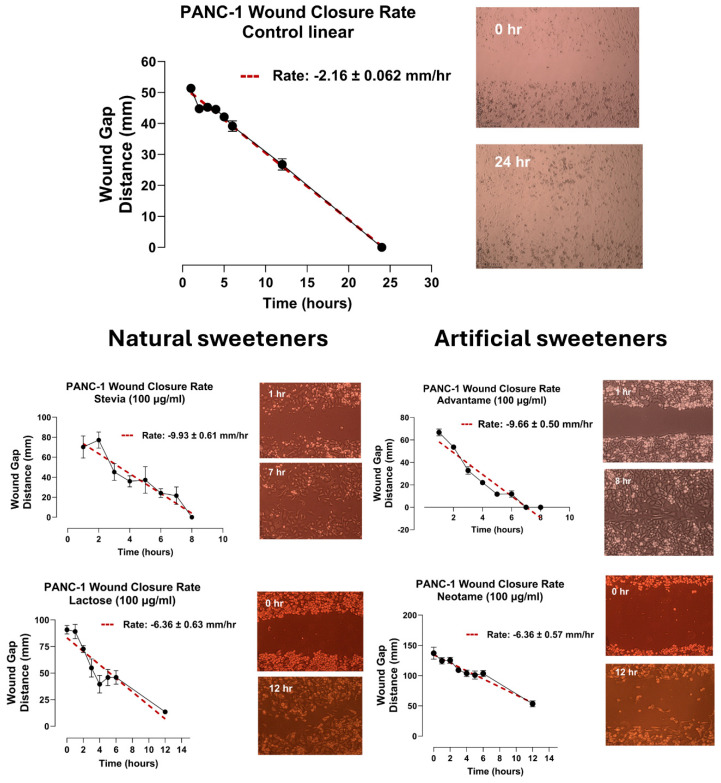
Natural (stevia and lactose) and artificial (advantame and neotame) sweeteners enhance the migratory potential of PANC-1 cancer cells using the scratch wound assay. The wound closure rate of untreated PANC-1 cells was over 24 h. The best-fit slope represents the rate in mm/hr of wound closure.

**Figure 10 nutrients-16-01840-f010:**
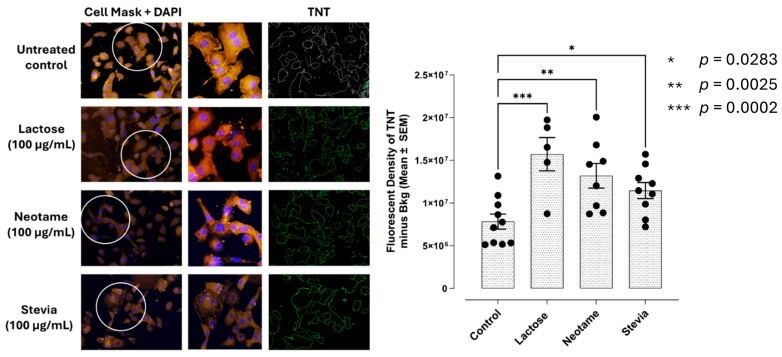
Neotame, lactose and stevia sweeteners induce tunneling nanotubes (TNT) in PANC-1 cells.

**Figure 11 nutrients-16-01840-f011:**
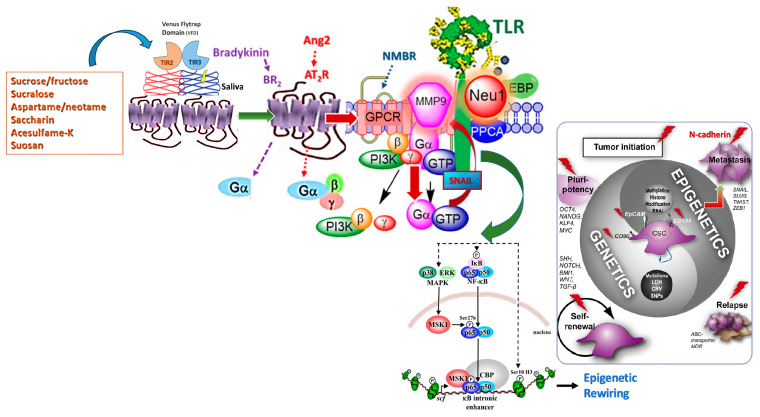
Dimeric taste receptor T1R2/T1R3 participates in a multimeric receptor complex with neuromedin B receptor (NMBR), neuraminidase 1 (Neu-1), and Toll-like receptor (TLR). Here, the biased functional T1R2/T1R3 GPCR oligomeric platform potentiates Neu-1 and MMP-9 cell surface crosstalk, mediating TLR glycosylation modification and transactivation and subsequent NF-κB-induced epigenetic rewiring. Artificial sweeteners binding to T1R2/T1R3 GPCR are illustrated. Notes: Sweetener stimulation of the biased T1R2/T1R3 GPCR induces receptor heterodimerization with NMBR, wherein NMBR-induced activation of SNAIL to induce MMP-9 endopeptidase to cleave the elastin binding protein (EBP) and expose the catalytic sialidase domain of Neu-1. The sialidase domain of Neu-1 hydrolyzes α-2,3-sialic acid from the glycosylated TLR receptor, reducing the steric hindrance and facilitating TLR dimerization, activation, and downstream cellular signaling. The resultant downstream signaling mediates the phosphorylation of the IkB subunit, which facilitates the translocation of NF-κB to the nucleus, enabling epigenetic modulation of gene expression [28,44]. Citation: Reprinted/Adapted with permission Jakowiecki et al. [45] Molecules 2021, 26, 2456, Publisher and licensee MDPI, Reber et al. [43] PLoS One 2009, 4, e4393, Publisher and licensee PLOS and Marquardt et al. [46] J Hepatol 2010, 53, 568-577, Publisher and licensee Elsevier. These articles are open-access articles distributed under the terms and conditions of the Creative Commons Attribution (CCBY) license (http://creativecommons.org/licenses/by/4.0/ (accessed on 23 April 2021)), It permits unrestricted use, distribution, and reproduction in any medium, provided the original author and source are properly credited.

## Data Availability

All data needed to evaluate the paper’s conclusions are present. The preclinical datasets generated and analyzed during the current study are not publicly available but are available from the corresponding author upon reasonable request. The data will be provided following the review and approval of a research proposal, Statistical Analysis Plan, and execution of a Data Sharing Agreement. The data will be accessible for twelve months for approved requests, considering possible extensions; contact szewczuk@queensu.ca for more information on the process or to submit a request.

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
