# Peer review of "Artificial and Natural Sweeteners Biased T1R2/T1R3 Taste Receptors Transactivate Glycosylated Receptors on Cancer Cells to Induce Epithelial–Mesenchymal Transition of Metastatic Phenotype"

_nutrients, 2024, doi:10.3390/nu16121840_

Round 1

Reviewer 1 Report

Comments and Suggestions for Authors

The paper “Artificial and Natural Sweeteners Biased T1R2/T1R3 Taste Receptors Transactivate Glycosylated Receptors on Cancer Cells to Induce Epithelial–Mesenchymal Transition of Metastatic Phenotype” presents a novel molecular mechanism of how NNS and natural sugars binding taste receptors transmogrify glycosylated receptors on cancer cells to induce the epithelial-mesenchymal transition of the metastatic phenotype. Here are some questions that need to be further improved or explained.

Comments:

Q1. “Non-nutritive sweeteners (NNS) are widely used by individuals to sustain a healthy diet” and “NNS consumption has been associated with increased risk factors for metabolic syndrome (MetSyn), resulting in diseases like cardiovascular dysfunction” are contradictory. Besides, other similar contradictory expressions in the article need to be further revised.

Q2. The repetition rate of articles is too high, over 50%, to be acceptable.

Q3. The same reference has been cited repeatedly in the article for many times. It is suggested to delete some citations or simplify the description of the article.

Q4. The cell photograph in Figure 1 cannot be seen clearly.

Q5. The article contains numerous lengthy paragraphs. It is recommended that additional summaries would effectively decrease the repetition rate.

Q6. As for the definition of * in the article, it is recommended to refer to other literatures, and generally it should not represent p equal to a certain value.

Q7. The data presented in this paper are sufficient; however, there is a lack of consistency in the main line, meaning that the raw materials utilized for the majority of indicators should be uniform rather than employing different drug interventions in separate experiments.

Q8. Is there any evidence that the sweeteners used in this article could reach the cell surface directly after consumption without being metabolized by the body?

Author Response

The paper “Artificial and Natural Sweeteners Biased T1R2/T1R3 Taste Receptors Transactivate Glycosylated Receptors on Cancer Cells to Induce Epithelial-Mesenchymal Transition of Metastatic Phenotype” presents a novel molecular mechanism of how NNS and natural sugars binding taste receptors transmogrify glycosylated receptors on cancer cells to induce the epithelial-mesenchymal transition of the metastatic phenotype. Here are some questions that need to be further improved or explained.

Comments:

Q1. “Non-nutritive sweeteners (NNS) are widely used by individuals to sustain a healthy diet” and “NNS consumption has been associated with increased risk factors for metabolic syndrome (MetSyn), resulting in diseases like cardiovascular dysfunction” are contradictory. Besides, other similar contradictory expressions in the article need to be further revised.

Author response: Thank you for this comment. We have removed the first sentence in the introduction as pointed out here. We have checked other areas in the manuscript as well. Most of the text is conceptual concepts taken from the literature to support our rationale of the study.

Q2. The repetition rate of articles is too high, over 50%, to be acceptable.

Author response: Thank you for this comment. We have checked repetitions and removed them.

Q3. The same reference has been cited repeatedly in the article many times. It is suggested to delete some citations or simplify the description of the article.

Author response: Thank you for this comment. We have checked repetitions and removed them.

Q4. The cell photograph in Figure 1 cannot be seen clearly.

Author response: Thank you for this comment.  Figure 1B does reveal the relative cell density on the slides for all of the sialidase assays. The sialidase assay only shows the fluorescent in blue image. The assay measures activated Neu-1 which hydrolyzes the 4-MUNANA sialidase substrate surrounding live cells, to form free 4-methylumbelliferone (4-MU), which fluoresces at 450 nm (blue color) when excited at 365 nm.  

We have published the protocol of the sialidase assay:

Amith, Jayanth et al. (2010) JoVE_Protocol_2142 (DOI 10.37912142)

Q5. The article contains numerous lengthy paragraphs. It is recommended that additional summaries would effectively decrease the repetition rate.

Author response: Thank you for this comment. We have checked some of the lengthy paragraphs and removed them.

Q6. As for the definition of * in the article, it is recommended to refer to other literatures, and generally it should not represent p equal to a certain value.

Author response: Thank you for this comment. All figures have the p values embedded within the figure.

Q7. The data presented in this paper are sufficient; however, there is a lack of consistency in the main line, meaning that the raw materials utilized for the majority of indicators should be uniform rather than employing different drug interventions in separate experiments.

Author response: Thank you for this comment. The data using the different sweeteners including artificial and natural ones were based on the interesting results obtained. Noteworthy, sweet-sensing selectivity is mediated by different oligomerizations of related GPCRs on the cell surface. When sweet taste-sensing receptors are stimulated by one of four different sweeteners, including sucralose, acesulfame potassium, sodium saccharin, or glycyrrhizin, distinct signaling pathways are activated in pancreatic ß-cells [3]. Patterns of cytoplasmic Ca2+ and cAMP-induced changes by these sweeteners were all different from each other. Here, the data support the concept that sweeteners are biased agonists to activate sweet taste-sensing receptors. It was important to test different artificial and natural sweeteners in the study based on this functional selectivity taste GPCRs. Also, we used specific inhibitors as previously published by us (Bunsick et al. 2024) to illustrate the oligomerizations of related taste GPCRs with the RTK and TLR on the cell surface.

[3] Kojima, I.; Nakagawa, Y.; Ohtsu, Y.; Medina, A.; Nagasawa, M. Sweet Taste-Sensing Receptors Expressed in Pancreatic beta-Cells: Sweet Molecules Act as Biased Agonists. Endocrinol Metab (Seoul) 2014, 29, 12-19, doi:10.3803/EnM.2014.29.1.12.

David A. Bunsick 1, Jenna Matsukubo 1,2, Rashelle Aldbai 1, Leili Baghaie 1 and Myron R. Szewczuk 1,*. Functional Selectivity of Cannabinoid Type 1 G Protein-Coupled Receptor Agonists in Transactivating Glycosylated Receptors on Cancer Cells to Induce Epithelial–Mesenchymal Transition Metastatic Phenotype. Cells 2024, 13, 480. https://doi.org/10.3390/cells13060480.

Q8. Is there any evidence that the sweeteners used in this article could reach the cell surface directly after consumption without being metabolized by the body?

Author response: Thank you for this comment. A recent study by investigators at the National Institute of Diabetes, Digestive and Kidney Diseases at the National Institutes of Health measured how much artificial sweetener is absorbed into the blood stream by children and adults after drinking a can of diet soda. Results of this study are published in Toxicological & Environmental Chemistry. The team measured the artificial sweeteners sucralose and acesulfame-potassium, which are found in a wide range of packaged foods and beverages. These artificial sweeteners, also including saccharin and aspartame, have received a lot of attention lately because it has been found that they are not inert chemicals with a sweet taste, but active substances that can affect the metabolism. Despite their approval as food additives following the submission of detailed safety data to the United States Food and Drug Administration (FDA), concerns about their safety and especially about their long-term health effects remain.

Given this background, the authors performed a study to extend previous investigations into plasma concentrations of sucralose and acesulfame-potassium. Artificial sweetener concentrations were measured among adults following ingestion of various doses of sucralose with or without acesulfame-potassium, both in diet soda and mixed in seltzer or plain water. Results obtained in adults were then compared with measurements obtained in children. The study comprised 22 adults aged 18-45 and 11 children aged 6-12 with no known medical conditions and who were not using any medications, enrolled in a randomized same-subject crossover study. The protocol was approved by the Institutional Review Board of the National Institute of Diabetes and Digestive and Kidney Diseases (NIDDK). The results of the study demonstrated that, compared to adults, children had double the concentrations of plasma sucralose after ingestion of a single twelve ounce can diet soda. The same research team previously found that these artificial sweeteners were also present in breast milk when mothers ingested foods, drinks, medicines, or other products that contained artificial sweeteners.

This important question proposes a new study using artificial and natural sweeteners in preclinical animal models of pancreatic cancer. This is in progress.

Reviewer 2 Report

Comments and Suggestions for Authors

The manuscript titled Artificial and Natural sweeteners biased T1R2/T1R3 taste receptors transactivate glycosylated receptorson cancer cells to induce epithelial-meshenchymal transition of metastatic phenotype by Skapinker et al is an intriguing and well-written in vitro study. The study is good, well-written, but the presence of discussion within the results is distracting. The result section must be rewritten to present only the present results. How the study adds to the literature should be confined to the discussion.

Comments on the Quality of English Language

English use is appropriate and may require minor editing.

Author Response

The manuscript titled Artificial and Natural sweeteners biased T1R2/T1R3 taste receptors transactivate glycosylated receptors on cancer cells to induce epithelial-mesenchymal transition of metastatic phenotype by Skapinker et al is an intriguing and well-written in vitro study. The study is good, well-written, but the presence of discussion within the results is distracting. The result section must be rewritten to present only the present results. How the study adds to the literature should be confined to the discussion.

Author response: Thank you for this comment. We have revised the results as requested.

Round 2

Reviewer 1 Report

Comments and Suggestions for Authors

No additional comments.

Author Response

No additional comments.

Thank you for your support.

Reviewer 2 Report

Comments and Suggestions for Authors

The results section of the revised manuscript is still filled with text other than results. For each section there is an introduction paragraph--this material is (or should be provided) in the introduction. Following the presentation of the results, there is a statement of interpretation of the results --this is discussion. Further the figure legend contains methods that are not always (but should be) included in the methods section and not as a part of the figure legend. There are sections in the manuscript which separate the introduction, methods, results, and discussion--this manuscript does not adhere.

Comments on the Quality of English Language

The English use is acceptable.

Author Response

The results section of the revised manuscript is still filled with text other than results. For each section there is an introduction paragraph--this material is (or should be provided) in the introduction. Following the presentation of the results, there is a statement of interpretation of the results --this is discussion. Further the figure legend contains methods that are not always (but should be) included in the methods section and not as a part of the figure legend. There are sections in the manuscript which separate the introduction, methods, results, and discussion--this manuscript does not adhere.

Author response: Thank you for your comments. We have reviewed other articles published in this journal, and have revised the manuscript according to this format and the recommendation by this reviewer. We thank you for your time and cooperation.

Round 3

Reviewer 2 Report

Comments and Suggestions for Authors

The revised manuscript is much improved and I now find it acceptable. Thank you.

Comments on the Quality of English Language

English use is appropriate with possible minor editing.

Author Response

The revised manuscript is much improved and I now find it acceptable. Thank you.

Thank you for the review.